cognition/psychology

maths, anxiety, education

**Author for correspondence:**
Andy P. Field
e-mail: andyf@sussex.ac.uk

# Predicting maths anxiety from mathematical achievement across the transition from primary to secondary education

Andy P. Field[1], Danielle Evans[1], Tomasz Bloniewski[2,3] and Yulia Kovas[2,3]

[1]School of Psychology, University of Sussex, Brighton, UK
[2]Department of Psychology, Goldsmiths, University of London, London, UK
[3]International Centre for Research in Human Development, Tomsk State University, Tomsk, Russia

APF, 0000-0003-3306-4695; DE, 0000-0002-5330-3393

The primary- to secondary-education transition is a substantive life event for many children. The transition can be associated with changes in the developmental trajectories of both emotional health and academic achievement. The current study looked at whether the trajectory of mathematical attainment and emotional health (anxiety) across this transition predicted later maths anxiety. A secondary analysis of data from the Twin Early Development Study (TEDS) was performed. The statistical models were fit on the 753 participants (one from each twin pair) for which there were measures of mathematical performance across the primary- to secondary-education transition and maths anxiety at age 18. Two multi-level growth models were fit predicting mathematical attainment and anxiety over the primary- to secondary-education transition. The intercepts and slopes for each child were extracted from these models and used as predictors of subsequent maths anxiety at age 18. These effects were adjusted for biological sex, socio-economic status, verbal cognitive ability and general anxiety. Maths anxiety at age 18 was significantly predicted by both pre-transition levels of anxiety and mathematical attainment and their rate of change across the primary- to secondary-education transition. However, the effects were small, suggesting that theories of maths anxiety may have overplayed the role of prior mathematical attainment and general anxiety.

# 1. Introduction

Poor maths achievement and abilities in childhood have far-reaching consequences leading well into adulthood [1]. Numeracy difficulties in childhood costs over £760 million per year to the public in the UK alone, with poor numeracy and literacy combined costing over £2.3 billion [2]. Failing to address these problems early in development leads to greater unemployment, lower income, fewer promotions, school exclusions, substance misuse, depression and criminal activity [2–4]. Poor maths skills further limit educational opportunities, for example, around 49% of working-age adults have the maths skills expected of primary-school children, with only 22% of working-age adults having the equivalent of a C-grade or above in GCSE maths, which is widely required for additional education courses in the UK [5]. The cost to the public, and the individuals affected (i.e. through poor health and low income) demonstrates the importance of understanding the causes of low achievement. Previous research has predominantly investigated underlying cognitive abilities (e.g. IQ, working memory and processing speed); however, recent research has reviewed and highlighted the importance of affect within the performance and development of maths abilities, particularly focusing on the effects of maths anxiety [6,7].

Maths anxiety is defined as fear, tension and discomfort which are felt by some individuals in situations involving mathematics, which may interfere with the performance of mathematical tasks [8]. Maths anxiety can elicit severe emotional reactions to maths tasks [8], and has similar symptoms to those that contribute to the diagnostic criteria of anxiety disorders. Despite a moderate correlation between maths anxiety and general anxiety, $r = 0.35$ [9], and despite partial aetiological overlap with other anxiety types [10], maths anxiety is widely recognized as a distinct construct [11]. The estimated prevalence of maths anxiety varies significantly across studies, partly due to differences in both measurement and the operationalization of diagnostic criteria [7]. Chinn reported estimates of 2–6% for high maths anxiety in secondary school pupils in the UK [12]. Similarly, a large study of 1757 UK school children between years 4 (age 8–9) and 7 (age 12–13) found that 11% of children had an average score above 'moderate anxiety' on a maths anxiety scale [13]. Researchers investigating less severe levels of maths anxiety (mostly in students) have tended to report higher estimates, between 11% to 68% [14–17]. Regardless of the variability in these estimates, there are evidently substantial numbers of individuals experiencing varying levels of maths anxiety, and its corresponding negative impact.

Maths anxiety is not restricted to academic environments, it is experienced in and interferes with a wide range of everyday situations [18]. It has been linked to poor financial planning [19], low self-efficacy in teachers [20] and poor drug calculations in nurses [21]. In academic settings, maths anxiety is a barrier to learning related subjects such as statistics [22,23], avoidance of optional maths courses within education [8], and the avoidance of maths and science, technology, engineering and maths (STEM)-related careers [1,8,24]. This avoidance of maths perpetuates a negative cycle. For example, people experiencing maths anxiety are more likely to avoid maths-related situations, leading to fewer opportunities to practise and develop their skills, resulting in worse performance later on and consequently greater levels of maths anxiety [8].

A relationship between maths anxiety and poor maths performance has been consistently found in a range of studies [7,14,25–30], and has been identified in children as young as 6 and 7 years old [31,32]. Two meta-analyses report the correlation between maths anxiety and maths performance to be between $r = -0.27$ to $-0.36$ in high school and college students [27,29]. The most recent meta-analysis reports the overall correlation as $r = -0.27$ [$-0.31, -0.23$] in primary school children and $r = -0.36$ [$-0.40, -0.32$] in secondary school children [33]. Other studies have reported correlations of $r = -0.35$ in college students [25,26], with evidence of a stronger relationship between high school grades and maths anxiety of $r = -0.67$ [25]. Furthermore, maths anxiety is strongly related to maths enjoyment, $r = -0.75$, and the motivation to do well in maths, $r = -0.64$ [27]. These effects of anxiety are limited to maths performance with very little influence on verbal tasks [7].

The detrimental relationship between maths anxiety and poor maths achievement is a universal problem. Results from the 2012 Programme for International Student Assessment (PISA) show that on average across participating Organisation for Economic Co-operation and Development (OECD) countries, higher maths anxiety is associated with a 34-point lower score (on average) in mathematics, reportedly the equivalent of almost one year of schooling [34]. The same report suggests that the number of students experiencing maths anxiety increased slightly from 2003 to 2012, with the most significant increases for students in New Zealand, Sweden, Uruguay, Australia, Tunisia and Norway. The widespread effects of maths anxiety highlight the importance of research in this area. Children from various countries and cultures are struggling within their maths education, so much that they are potentially 'losing' a year or more of schooling—meaning early intervention is vital. However, the

success of any potential intervention strategy relies heavily upon the current understanding of the underlying causes and issues associated with increased maths anxiety.

To sum up, maths anxiety is a widespread problem with far-reaching negative consequences for school leavers both in everyday life and if they pursue further education. As such, understanding what predicts maths anxiety is important to better inform attempts to mitigate those negative impacts. Logically, there are three plausible descriptions of the maths anxiety–maths performance relationship. First, maths anxiety could cause poorer maths performance by impairing the processing and retrieval of information and fostering avoidance (and therefore practice) of maths tasks. This possibility has been labelled the the debilitating anxiety model [35]. Second, the reverse could be true: the deficit model describes the possibility that poor maths performance leads to maths anxiety [35]. The final possibility is that the relationship between maths anxiety and maths performance is bidirectional (the reciprocal model). Within this final model anxiety is seen to impair performance, but this poor performance is believed to increase anxiety in subsequent maths activities [35].

Dissociating the debilitating anxiety model from the deficit model hinges on demonstrating whether maths anxiety precedes poorer performance or vice versa. There is direct evidence that maths anxiety can predict poorer performance from studies that have manipulated anxiety levels and shown a corresponding change in subsequent maths performance [36,37]. There is also indirect evidence in that individuals experiencing maths anxiety are more likely to avoid maths tasks, which implies that maths anxiety leads to fewer opportunities to practise maths skills and, therefore, improve performance [8,27]. However, there is also evidence that poorer performance can precede maths anxiety. For example, longitudinal studies have shown that the correlation between academic performance in one year and maths anxiety in the following year is stronger than the correlations between maths anxiety in one year and academic performance in the following year [28]. On the basis that existing research cannot dissociate the debilitating anxiety model from the deficit model it is reasonable to assume that the reciprocal model is, at present, the most plausible description of the maths anxiety–maths performance relationship. Given that, the current study aims to look at predictors of maths anxiety at school-leaving age (18).

To date, several sources of maths anxiety have been identified including genetic and environmental factors [6,38]. Genetic risk factors account for up to 40% of the variance in maths anxiety [10,39]. The remaining variance will be explained by direct environmental influences and their interactions with genetic risk. For example, the expression of a genetic risk for maths anxiety is likely to be influenced by aspects of the home environment such as parental involvement [40], and the school environment, such as teachers' maths anxiety and its knock-on effect on their mathematical and teaching ability [41]. As already discussed, prior maths performance seems likely to predict future maths anxiety. Maths anxiety may stem from very early deficits in numerical abilities [42–44] such that individuals with poor basic maths abilities when starting school are more likely to experience maths anxiety later on [45]. The first hypothesis is, therefore, that early mathematical performance will predict maths anxiety at age 18 longitudinally. Also, the trajectory of a child's mathematical attainment should predict future maths anxiety: those with a slower rate of change in their mathematical attainment will have greater maths anxiety in the future.

As mentioned before, 60% of the variance in maths anxiety can be attributed to environmental influences and their interactions with genetic risk. These influences might act on maths anxiety through their influence on the child's mathematical attainment and learning trajectory, but also through their influence on the child's emotional well-being. It is likely that characteristics of the individual feed into both maths anxiety and maths performance and are themselves influenced by the school environment and how it is perceived. Psychological theories such as the attentional control theory provide a plausible underlying mechanism through which anxiety impairs performance [46,47]. The main assumption in this theory is that anxiety experienced in the moment (state anxiety) increases attentional allocation towards threat-related stimuli such as worrisome thoughts, resulting in less attention available for performance-based tasks. The system involved in allocating attention is the central executive and other executive functions including working memory, shifting, updating and inhibition [48,49], which have been found to be particularly important in the development and performance of maths abilities [50]. Additionally, maths is arguably more impaired by anxiety compared to other subjects because maths is particularly taxing on executive function skills [51]. For example, in mental arithmetic tasks individuals are required to use their working memory, as well as shifting and updating [52,53] to be successful. Therefore, if the experience of anxiety takes up a large amount of the executive function allocation, less allocation is available for working memory, and the shifting and updating processes. Working memory has been found to correlate with maths anxiety [25] and performance is particularly impaired by maths anxiety on high working-memory load tasks [26].

The effects of state anxiety on performance are moderated by an individual's dispositional tendency to experience anxiety (trait anxiety). The processing biases associated with state anxiety are activated at

lower levels of 'threat' in individuals with high trait anxiety compared to those with low trait anxiety [54]. Put another way, the effects of state anxiety are exacerbated in those high on trait anxiety. As such, high levels of trait anxiety should predict maths anxiety but this relationship should be exacerbated by elevated levels of state anxiety. The school environment has the potential to create anxiety at school and this state anxiety, which has a knock-on effect on mathematical performance, will be heightened for trait anxious children. The current study focuses on one particularly stressful era in many children's educational journey: the transition from primary to secondary education.

The school environment is a large component of the child-specific environment, and during the primary–secondary education transition the school environment changes radically for many children (for a review see [55]). For example, in the UK, in primary school children typically experience a smaller, more personalized class taught by a single teacher and learning is task-oriented. In contrast, secondary schools typically have larger, more impersonal, classes taught by multiple teachers and learning is performance-focused. This transition, when children leave their primary education institution to attend a typically separate secondary education institution usually between the ages of 10 and 14 years, is potentially highly influential in creating or exacerbating maths anxiety. The primary to secondary education transition is a normative event for most children that is widely recognized as one of the most stressful events young adolescents will experience [56,57]. It can elicit various negative academic, behavioural and emotional outcomes [55]. The transition coincides with pubertal development in adolescents and is also a period where emotional disorders become more salient [58]. Increases in general anxiety and school concerns around the time of transition have also been widely reported [59–63]. The transition is a potentially problematic time for young adolescents and the negative effects can be extensive. In addition to declines in general academic achievement [e.g. 59,64], maths achievement is stifled during the transitioning year with around 34% of pupils making no progress in maths between primary and secondary education [65]. Furthermore, interest, enjoyment, confidence and self-efficacy in maths decline following the move to secondary school [66–69]. Maths anxiety has been found to increase at the time of the transition for students that moved to a new secondary school compared to those that did not, with greater increases reported for females and high-achievers [70]. Additionally, Hill *et al.* [71] found a significant association between high maths anxiety and low maths performance in secondary education students, but not for those in primary education. The secondary education pupils in their sample had recently transitioned to their new school, raising the question of whether the transition event itself impacted the relationship found between maths anxiety and maths achievement.

To sum up, there is likely to be a reciprocal relationship between maths anxiety and maths performance. Maths performance early in a child's academic journey is likely to predict future maths anxiety. Compared to children whose early performance is strong and whose educational trajectory over time follows the expected course, those with poorer early maths performance, or whose trajectory over time falls short of expectations, would be expected to experience more future maths anxiety. The trajectory of a child's maths performance is likely to be affected by both the school environment, the individual characteristics of the child, and by the interplay between the two. With respect to the school environment, the transition from primary to secondary education represents a significant change in the environment that has a negative impact on many children. In addition, the impact that it has is likely to be worse for children high on trait anxiety. Specifically, it might heighten or maintain high levels of trait anxiety. Therefore, trait anxiety before and after the transition from primary to secondary education should also predict future maths anxiety.

This study aims to predict maths anxiety longitudinally using a secondary analysis of the Twin Early Development Study (TEDS). It is hypothesized that poor maths achievement during the primary–secondary education transition will predict later maths anxiety. In addition, it is predicted that children experiencing increased emotional difficulties before the transition will be at a greater risk of a negative transition to secondary education, leading to a poorer maths trajectory across the transition and increased maths anxiety at age 18.

# 2. Method

## 2.1. Sample

This study was a secondary data analysis of the Twins Early Development Study (TEDS), a UK longitudinal twin registry that recruited from all families living in England and Wales who had twin-births between 1994 and 1996. The sample comprises four birth cohorts and the current analysis included participants recruited

from the first two cohorts, aged between 18 and 21. The initial cohort comprised 12 586 children, but only 1457 of those completed maths anxiety measures at age 18. Of these children, maths attainment information across the school transition was available for only 1104. The purpose of the paper was not to produce genetically informed models, therefore only one randomly selected member of each twin contributed to the models. The sample is representative of the UK population [72].

## 2.2. Measures

### 2.2.1. Maths anxiety at age 18

Maths anxiety was assessed using a modified version of the Abbreviated Math Anxiety Scale, AMAS [73]. The AMAS includes nine descriptions of maths-related activities to which participants indicate their anxiety on a 5-point scale ranging from 1 = *not nervous at all* to 5 = *very nervous*. Some examples are 'Reading a maths book' and 'Listening to a maths lecture'. Some items were modified to make the scale age appropriate (for details see [10]). The average response was used yielding a score that could range from 1 to 5. The AMAS has excellent internal validity, $\alpha = 0.90$ [73]. The modified AMAS used in the TEDS sample also has excellent internal validity, $\alpha = 0.94$, and test–retest reliability, $r = 0.85$ [10].

### 2.2.2. General anxiety at age 18

General anxiety was measured using an online version of the 7-item Generalized Anxiety Disorder Scale, GAD-7, which is a validated and reliable measure of anxiety in the general population [74]. For each of seven problems, participants rated how often they have been bothered by it in the past two weeks on a 4-point scale (1 = *not at all*, 4 = *nearly every day*). Example problems are: 'Not being able to control worrying', 'Have trouble relaxing' and 'Feeling afraid as if something awful might happen'. Responses are scored from 0 to 3 resulting in a total score that can range from 0 to 21. The GAD-7 has good internal consistency, $\alpha = 0.89$ and test–retest reliability, $r = 0.64$, generally [74], and good internal consistency within the TEDS sample, $\alpha = 0.91$ [10].

### 2.2.3. Maths attainment

Maths achievement was measured pre- and post-transition to secondary education at ages 9 and 12 using teacher ratings based on the UK National Curriculum (NC) assessment guidelines followed by teachers within the UK state school system. Teachers were contacted in the second half of the school year to ensure they were knowledgeable about the child's maths performance. The scores were a composite of ratings of the following mathematical skills: using and applying mathematics; number and algebra; shape, space and measures. Raw NC ratings can range from 1 to 9, but reflect age-related curriculum content so, for example, a child aged 9 is highly unlikely to have the relevant skills to be rated as a 9 (at this age a typical score would be 4). The National Curriculum assessments have been shown to be valid measures of academic achievement [75,76]. Further information about the National Curriculum is available at https://www.gov.uk/national-curriculum.

### 2.2.4. Verbal attainment

Participants at age 10 and 12 were tested on a web-based adaptation of the WISC-III Multiple Choice Information (General Knowledge) and WISC-III Vocabulary Multiple Choice [77,78]. The tests at age 12 were the same tests used at age 10 but with the addition of more difficult age-appropriate items. For a full description of the measures see [79], but in brief the Vocabulary Multiple Choice consisted of 30 vocabulary questions (e.g. what does 'migrate' mean?) and the Multiple Choice Information contained 30 general knowledge questions (e.g. in which direction does the sun set?). For both scales children select one of three or four possible responses for each item. The scores were combined as the mean score standardized on a filtered sample in which all exclusions are removed using appropriate standardized web test scores. In the current sample, internal consistency is high for both vocabulary, $\alpha = 0.90$ (age 10) and 0.88 (age 12), and general knowledge, $\alpha = 0.87$ (age 10) and 0.81 (age 12) [79].

### 2.2.5. Anxiety across the school transition

General anxiety levels at ages 9 and 12 were assessed using the emotional symptoms subscales of the Strengths and Difficulties Questionnaire, SDQ [80,81]. The SDQ contains 25 items that decompose into

**Table 1.** Key model parameters for multi-level growth models of maths attainment and SDQ over time.

| effect | b | 95% CI | t | d.f. | p |
|---|---|---|---|---|---|
| maths model | | | | | |
| intercept | 2.93 | [2.90, 2.95] | 237.75 | 5504.00 | 0.00 |
| time | 0.48 | [0.47, 0.50] | 81.75 | 981.00 | 0.00 |
| SDQ model | | | | | |
| intercept | 3.24 | [3.16, 3.32] | 80.67 | 6217.00 | 0.00 |
| time | −0.37 | [−0.39, −0.34] | −26.20 | 2586.00 | 0.00 |

SDQ, Strengths and Difficulties Questionnaire (anxiety subscale).

five factors, two of which broadly reflect externalizing symptoms (hyperactivity, conduct problems), two of which reflect internalizing problems (emotional symptoms, peer problems) and one of which reflects prosocial behaviour. Anxiety levels across the school transition were indicated by the five items from the emotional symptoms subscale, which relate to symptoms of anxiety (e.g. 'I worry a lot', 'I am nervous in new situations', 'I easily lose confidence', 'I have many fears, I am easily scared'). Children responded to items using a response scale of *not true*, *quite true* and *very true*, scored as 0, 1 and 2 respectively. As such, the total score at each time point could range from 0 to 10. The SDQ has good concurrent and predictive validity [80,81]. In community samples the internal consistency for the self-report version of the scale overall is high, $\alpha = 0.80$, and for the emotional problems subscale is acceptable, $\alpha = 0.66$ [80].

### 2.2.6. Socio-economic status

Within TEDS, the mother's and father's highest educational qualification and job status were collected at first contact with the families (i.e. 18 months). The socio-economic status (SES) index used in the current study was the mean of five standardized SES ratings: mother and father qualification level, mother and father occupational status and the mum's age at the birth of their first child.

### 2.3. Data analysis plan

Analyses were conducted using R v. 3.6.1 (2019-07-05) [82] and the following packages: *gmodels* [83], *mice* [84], *MissMech* [85], *nlme* [86] and *tidyverse* [87]. In phase one, two multi-level models were fit to SDQ emotional symptoms and maths attainment scores respectively. Both models had the following general form:

$$\text{Outcome}_{ij} = [\gamma_{00} + \gamma_{10}\,\text{Time}_{ij}] + [\zeta_{0i} + \zeta_{1i}\,\text{Time}_{ij} + \epsilon_{ij}].$$

In other words, SDQ emotional symptoms or maths attainment were nested within participants at different timepoints (9 and 12 years old). The intercept and slope for the effect of time were modelled as random effects. Table 1 shows the model parameters for these two models. Having fitted each model, the individual intercept and slope for time for each child was saved into the data. This process resulted in four new variables being created. The first two represented the intercept for SDQ emotional symptoms and maths attainment (i.e. the values at age 9) and the remaining two represented the slopes of SDQ emotional symptoms and maths attainment over time (i.e. the change from 9 to 12 years old).

In phase two, the main model was simply a linear model predicting maths anxiety at age 18 from SES, biological sex, verbal attainment, general anxiety at age 18 and the four variables from the previous models: SDQ emotional symptoms (pre-transition), SDQ emotional symptoms (change), maths attainment (pre-transition) and maths attainment (change). All predictors were entered simultaneously. Table 2 shows the correlations between these variables.

The starting sample size for the final models was $N = 1104$. Within this sample there were missing values across some of the predictors. There were 952 complete cases and the patterns of missing data are shown in the electronic supplementary material. Jamshidian and Jalal's nonparametric test of homogeneity of covariances was used to test whether data were missing completely at random (MCAR) [88]. If covariances are comparable (i.e. the test is not significant) across groups with different patterns of missingness then MCAR can be assumed. The test was non-significant, 0.36, and the paternal data 0.36

**Table 2.** Correlation matrix for variables in the models predicting maths anxiety (age 18).

| variable | M | s.d. | 1 | 2 | 3 | 4 | 5 | 6 | 7 |
|---|---|---|---|---|---|---|---|---|---|
| maths anxiety (age 18) | 2.28 | 1.00 | | | | | | | |
| general anxiety (age 18) | 1.96 | 0.74 | 0.35*** | | | | | | |
| SES | 0.32 | 0.95 | −0.07* | −0.05 | | | | | |
| maths (age 9) | 3.14 | 0.63 | −0.31*** | −0.09** | 0.28*** | | | | |
| maths (age 12) | 4.95 | 0.94 | −0.26** | −0.08* | 0.29*** | 0.53*** | | | |
| verbal attainment (age 10–12) | 0.20 | 0.89 | −0.19*** | −0.11*** | 0.34*** | 0.42*** | 0.43*** | | |
| SDQ anxiety (age 9) | 3.11 | 2.34 | 0.13*** | 0.17*** | −0.13*** | −0.15*** | −0.07 | −0.18*** | |
| SDQ anxiety (age 12) | 2.07 | 2.02 | 0.24*** | 0.26*** | −0.10** | −0.15*** | −0.09* | −0.15*** | 0.40*** |

SES, socio-economic status; SDQ, Strengths and Difficulties Questionnaire. *$p < 0.05$, **$p < 0.01$, ***$p < 0.001$.

**Table 3.** Summary statistics for the key study measures.

| measure | $n$ | min | max | *Mdn* | *M* | 95% CI | *s* |
|---|---|---|---|---|---|---|---|
| maths anxiety (age 18) | 1104 | 1.00 | 5.00 | 2.00 | 2.28 | [2.22, 2.34] | 1.00 |
| general anxiety (age 18) | 1104 | 1.00 | 4.00 | 1.71 | 1.96 | [1.91, 2.00] | 0.54 |
| maths (age 9) | 915 | 1.00 | 5.00 | 3.00 | 3.14 | [3.10, 3.18] | 0.40 |
| maths (age 12) | 609 | 1.33 | 9.00 | 5.00 | 4.95 | [4.87, 5.02] | 0.88 |
| SDQ anxiety (age 9) | 1006 | 0.00 | 10.00 | 3.00 | 3.11 | [2.97, 3.26] | 5.46 |
| SDQ anxiety (age 12) | 1015 | 0.00 | 10.00 | 2.00 | 2.07 | [1.95, 2.19] | 4.09 |
| SES | 1063 | −2.21 | 2.55 | 0.27 | 0.32 | [0.26, 0.38] | 0.90 |
| verbal attainment (age 10–12) | 1000 | −2.60 | 2.34 | 0.26 | 0.20 | [0.15, 0.26] | 0.79 |

SES, socio-economic status; SDQ, Strengths and Difficulties Questionnaire.

giving no reason to reject the assumption that data are missing completely at random. Missing data were handled using multiple imputation implemented with the mice [84] package using predictive mean matching. Seventy imputation samples were created and the final model was estimated by pooling the models fit to these samples [89,90].

# 3. Results

Table 3 shows summary statistics for all of the variables in the model. Scores on maths anxiety extended across the full range of the scale, but with a mean and median below the mid-point of the scale. SDQ emotional symptoms scores similarly covered the full range but were, on average, towards the low end with relatively high variance compared to the mean. Interestingly, emotional symptoms were lower post-transition compared to pre, indicating that anxiety had decreased. Maths performance increased from age 9 to 12 as would be expected based on how NC levels are scored.

Table 4 shows the model parameters for predictors of maths anxiety at age 18 (an equivalent table reporting the same model fitted to standardized scores can be found in the electronic supplementary material). With respect to statistical significance, biological sex, SDQ emotional symptoms (pre-transition), the change in SDQ emotional symptoms, maths attainment (pre-transition), the change in maths performance across the transition, and general anxiety at age 18 all significantly predicted maths anxiety. Bearing in mind that maths anxiety could range from 1 to 5, males were 0.37 points lower on this scale than females, and a point increase on the general anxiety scale (which ranged from 1 to 4) equated to a 0.36 increase in maths anxiety. With respect to the substantive predictors, a unit change in maths attainment pre-transition (at age 9 measured on a 5-point scale) equated to a 0.46 unit decrease in maths anxiety 8 years later at age 18. A unit increase in the rate of change (i.e. the slope) in maths attainment across the transition equated to a −0.32 decrease in maths anxiety at age 18. To unpick what this value means, first let's look at what a typical change in maths attainment would be across the school transition. In our earlier model we operationalized the primary-to-secondary education transition as the 3-year period between ages 9 and 12 and found that maths attainment increases at a rate of 0.48 units per year (table 1). A typical change in maths attainment across three years during which the school transition occurs would be $3 \times 0.48 = 1.44$ units. Imagine a child who shows no improvement in their maths ability over the same three years. Their slope for attainment will be 0, and it will be 1.44 units lower than a typical child. Their predicted maths anxiety at age 18 will correspondingly be $-1.44 \times -0.32 = 0.46$ higher on the 5-point maths anxiety scale than the average child. Put another way, the rate of change in maths attainment had a small effect on maths anxiety at age 18.

With respect to SDQ emotional symptoms, a unit change in pre-transition scores (at age 9 and measured on a 10-point scale) equated to a 0.06 unit increase in maths anxiety at age 18. To place this effect in perspective, a change equating to the entire length of the SDQ emotional symptoms scale predicts only a 0.63 increase in maths anxiety.

A unit increase in the change in SDQ emotional symptoms across the school transition equated to 0.21 unit increase in maths anxiety. The typical change in SDQ emotional symptoms across transition was

**Table 4.** Key model parameters for predictors of maths anxiety at age 18.

| predictor | b | 95% CI | t | d.f. | p |
|---|---|---|---|---|---|
| intercept | 3.19 | [2.77, 3.60] | 15.02 | 1076.18 | 0.00 |
| SES | 0.04 | [− 0.02, 0.10] | 1.40 | 985.64 | 0.16 |
| biological sex | −0.37 | [−0.48, −0.26] | −6.45 | 1090.74 | 0.00 |
| SDQ anxiety (pre-transition) | 0.06 | [0.03, 0.10] | 3.52 | 1028.91 | 0.00 |
| SDQ anxiety (change) | 0.21 | [0.09, 0.33] | 3.56 | 1024.95 | 0.00 |
| verbal attainment | −0.01 | [−0.09, 0.06] | −0.40 | 750.33 | 0.69 |
| maths (pre-transition) | −0.46 | [−0.57, −0.35] | −8.03 | 1067.01 | 0.00 |
| maths (change) | −0.32 | [−0.64, 0.00] | −1.96 | 1090.87 | 0.05 |
| general anxiety (age 18) | 0.36 | [0.28, 0.43] | 9.42 | 1088.99 | 0.00 |

SES, socio-economic status; SDQ, Strengths and Difficulties Questionnaire.

**Table 5.** Key model parameters for predictors of maths anxiety at age 18 (exploratory model).

| predictor | b | 95% CI | t | d.f. | p |
|---|---|---|---|---|---|
| intercept | 3.41 | [2.90, 3.92] | 13.19 | 1074.58 | 0.00 |
| SES | 0.04 | [−0.02, 0.10] | 1.38 | 979.77 | 0.17 |
| biological sex | −0.95 | [−1.71, −0.19] | −2.46 | 1086.29 | 0.01 |
| SDQ anxiety (pre-transition) | 0.08 | [0.03, 0.12] | 3.58 | 1009.39 | 0.00 |
| SDQ anxiety (change) | 0.24 | [0.10, 0.38] | 3.36 | 1008.82 | 0.00 |
| verbal attainment | −0.01 | [−0.09, 0.06] | −0.38 | 751.89 | 0.70 |
| maths (pre-transition) | −0.52 | [−0.67, −0.38] | −7.14 | 1070.35 | 0.00 |
| maths (change) | −0.44 | [−0.85, −0.02] | −2.08 | 1087.44 | 0.04 |
| general anxiety (age 18) | 0.36 | [0.28, 0.43] | 9.43 | 1084.84 | 0.00 |
| sex × maths (pre-transition) | 0.16 | [−0.05, 0.37] | 1.52 | 1088.13 | 0.13 |
| sex × maths (change) | 0.34 | [−0.32, 0.99] | 1.02 | 1088.02 | 0.31 |
| sex × SDQ (pre-transition) | −0.05 | [−0.12, 0.02] | −1.31 | 1064.05 | 0.19 |
| sex × SDQ (change) | −0.11 | [−0.35, 0.14] | −0.86 | 1067.53 | 0.39 |

SES, socio-economic status; SDQ, Strengths and Difficulties Questionnaire (anxiety subscale).

a change of − 0.37 units per year (i.e. anxiety decreased), or a $3 \times -0.37 = -1.11$ unit change across the 3-year school transition. Therefore, a decrease in SDQ emotional symptoms across the school transition that is typical in magnitude would equate to a change in maths anxiety of $-1.11 \times 0.21 = -0.23$ units on a 5-point scale. Again, a small reduction.

Table 5 shows an unplanned, *post hoc*, model that quantifies the degree to which biological sex moderated the effects on maths anxiety of maths attainment and SDQ emotional symptoms prior to and across the school transition. Biological sex did not significantly moderate any of the effects of maths attainment of emotional symptoms on subsequent maths anxiety. An equivalent table reporting the same model fitted to standardized scores can be found in the electronic supplementary material.

# 4. Discussion

This study shows that maths anxiety at age 18 was significantly predicted by biological sex and general anxiety (at age 18) but, most important, by maths attainment at age 9 prior to the transition to secondary education and by the change in maths attainment across that transition; and also both by anxiety symptoms on the SDQ prior to transition (also at age 9) and the change in anxiety symptoms across the transition.

The small effect of biological sex on maths anxiety at age 18 (males were 0.37 points lower on the maths anxiety scale) replicates the findings of previous research [7,70,91], suggesting that females experience increased maths anxiety throughout adolescence and adulthood. One proposed explanation is that females show higher levels of trait anxiety [92] and as a consequence the effects of state anxiety are exacerbated [54]. However, general anxiety was controlled for in the model, implying that sex differences in maths anxiety may develop from other aspects not investigated by the present study, such as stereotype threat [93] or a male advantage in maths performance at that age. As expected, greater general anxiety was found to predict increased maths anxiety at age 18 (a 1-point increase in general anxiety equated to a 0.36 increase in maths anxiety), consistent with wider literature [27] suggesting that general anxiety and maths anxiety share genetic and environmental risk factors [7,9,10,39].

The findings of this study also show that maths attainment pre-transition (at age 9), and the rate of change across the transition significantly predicted maths anxiety at age 18. For maths attainment pre-transition, 1 unit increase in attainment equated to a −0.46 unit change in maths anxiety, showing a moderate effect on maths anxiety at age 18. For the rate of change in maths attainment, a unit change in the slope of maths attainment equated to a −0.32 change in maths anxiety, which is a very small effect. Nonetheless, small though the effect is, it is consistent with the wider literature [7,27,28,30,42–44]. In this sense, the results support the hypothesis from the reciprocal and deficit models of maths anxiety that poor performance leads to increased maths anxiety, but also imply that this relationship is very weak. However, as maths anxiety was not assessed during the transition period (age 9–12), the current study says nothing about whether this relationship is reciprocal. Nevertheless, the rate of change in attainment significantly predicted maths anxiety (albeit a small effect), implying that those who do not transition successfully in terms of attainment, are more likely to experience negative outcomes several years later, even when adjusting for general anxiety. Ultimately, the small effect implies that there are other processes or mechanisms underlying the association that we did not explore in this study, including motivation, and interactions between motivation and anxiety. The genetic and environmental factors underlying these unexplored mechanisms may be the same or partly the same as those underlying the predictors that we did explore, but we cannot test this idea here.

The other main finding of this study was that greater anxiety symptoms pre-transition (age 9) as measured by the SDQ significantly predicted increased maths anxiety at age 18 ($b = 0.06$). The rate of change in anxiety symptoms also significantly predicted maths anxiety ($b = 0.21$), though when placing the size of the rate of change within the context of the typical change in anxiety seen between the ages of 9 and 12, the effect on maths anxiety at age 18 was very small (the change in maths anxiety equating to the typical change in anxiety symptoms was about a −0.23 shift on a 5-point scale). Because general anxiety at age 18 was adjusted for in the model, the effects of anxiety symptoms prior to and across the transition are unique to maths anxiety and not just predictors of general worries and anxiety symptoms. These findings support previous research that show correlations between general anxiety symptoms and maths anxiety [27,39,71], but shows that early anxiety significantly predicts future maths anxiety. The size of the effect found in previous studies has been much greater than that found in this study [27,71], but this could be explained by previous studies being cross-sectional, rather than looking at anxiety predicting maths anxiety some years later.

Overall, the findings suggest that a transition to secondary education characterized by poorer attainment and increased anxiety, both prior to and across the transition, has small but long-lasting consequences leading into adulthood. The secondary education environment is viewed as creating greater competitiveness and social comparisons between students which can elicit maths anxiety in children [55,94]. Combined with slower rates of attainment, and increased emotional difficulties during this time, students' self-confidence and self-efficacy could be easily negatively impacted, potentially creating the 'ideal' environment for maths anxiety to develop. While some researchers have found children as young as 6 and 7 years old have maths anxiety [31,32], it is possible that the transition to adolescence contributes to this process, given the changes in the learning environment, and the increasing difficulty of school work. The detrimental effect of high maths anxiety on careers, further study and general well-being as discussed previously, highlights the need to focus on this period in adolescence in terms of intervention strategies. However, the effects of emotional difficulties and maths attainment during this period were small, implying that the focus of future work should be broader than these variables alone.

Currently, schools employ various intervention strategies for children moving from primary to secondary education, which differ between institutions, and focus on improving self-confidence and problem-solving [95]. However, the small size of the effects also raises the question of whether it is beneficial to focus on improving attainment and reducing anxiety across the transition, *solely* with the goal to reduce maths anxiety later on, or whether there are more effective ways to decrease maths anxiety with larger gain.

There are several methodological limitations in the present study. Anxiety symptoms prior to and across the transition were generally low, meaning that the findings here may lack generalizability to individuals experiencing greater anxiety symptoms during pre-adolescence, or those having an extremely poor transition to secondary education in terms of emotional well-being. In addition, although the transition to secondary education is characterized by increased anxiety at the group level, in the current sample anxiety actually decreased during this period. This pattern of general anxiety may well explain why its effects on future maths anxiety were small. Perhaps this sample is atypical and future work might fruitfully look at samples from communities where the school transition is particularly problematic. Also, maths anxiety was not measured across the transition from primary to secondary education, meaning that it was not possible to adjust predictors for prior levels of maths anxiety during the transition, or look at any changes in trajectory when moving to secondary education to see if emotional and attainment issues during this period exacerbated maths anxiety itself. Finally, although the measures of maths attainment derived from teachers were based on the National Curriculum examination benchmarks, they nevertheless reflect subjective ratings of the children's ability and are not as objective as, for example, National Curriculum exam scores.

Overall, the findings suggest that children experiencing greater emotional difficulties and poor maths attainment prior to the transition to secondary education are more at risk of having higher maths anxiety at age 18. Furthermore, decreasing emotional difficulties and increasing attainment across the transition predicts lower maths anxiety at age 18. Given the small size of the effects, it is evident that there are other aspects that contribute to maths anxiety which should be the focus of future research. Nevertheless, transition strategies should remain focused on improving academic attainment (especially for female students) and emotional well-being, and be aware of the effects of a successful transition to secondary education on maths anxiety later in adulthood.

Ethics. This paper reports a secondary analysis of existing data and, therefore, required no ethical approval. Ethical approval for the TEDS study was received from King's College London Ethics Committee (Ref PNM/09/10-104); informed consent is always obtained prior to collecting data.

Data accessibility. Data used for this submission may be made available on request to the Twins Early Development Study (TEDS), through their data access mechanism (see http://www.teds.ac.uk/researchers/teds-data-access-policy). They will then consider requests for sharing data for appropriate research purposes. The R markdown file for this manuscript, which includes all R code for the reported analyses, is posted at https://osf.io/fp5xm/.

Authors' contributions. A.P.F., T.B. and Y.K. conceived the project. T.B. provided advice on measures and conducted initial data processing. A.P.F. ran all statistical analyses. A.P.F., D.E. and Y.K. wrote the manuscript. All authors gave final approval for publication.

Competing interests. We declare we have no competing interest.

Funding. We gratefully acknowledge the ongoing contribution of the participants in the Twins Early Development Study (TEDS) and their families. TEDS is supported by a programme grant to Robert Plomin from the UK Medical Research Council (MR/M021475/1 and previously G0901245), with additional support from the US National Institutes of Health (AG046938). The research leading to these results has also received funding from the European Research Council under the European Union's Seventh Framework Programme (FP7/2007-2013) grant agreement no. 602768 and ERC grant agreement no. 295366.

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
