## [Reviewer comments · Royal Society Open Science]

Review History

RSOS-191459.R0 (Original submission)

Review form: Reviewer 1

Is the manuscript scientifically sound in its present form?

Yes

Are the interpretations and conclusions justified by the results?

Yes

Is the language acceptable?

Yes

Do you have any ethical concerns with this paper?

No

Have you any concerns about statistical analyses in this paper?

No

Recommendation?

Accept as is

Comments to the Author(s)

A handful of recent papers have examined the issue of which construct, math anxiety or math achievement, leads to or influences the other, in an attempt to gain some evidence on causality on this knotty question. As the authors note, three theories have been articulated – the Debilitating Anxiety Model, the Deficit Model, and the Reciprocal Model – but all suffer from the same limitations, that the data brought to bear on the issue necessarily come from correlations, hence cannot support causal interpretations. The recent papers, on the other hand, have attempted to gain some leverage on the issue by examining patterns of influence (again, largely the magnitudes of correlations) in longitudinal designs, asking for example if poorer performance at time 1 leads to worse anxiety at time 2 or, instead whether worse anxiety at time 1 leads to worse performance at time 2.

The current manuscript examines the same kind of evidence, but I believe is considerably stronger than the recent papers I've seen, for a variety of reasons. First, it presents data from a far larger sample, well over 1000 cases, in contrast to the 100 or 200 cases normally tested. Second, it spans a considerably longer time range to examine the longitudinal effect; here the range of interest is prior to the transition from primary to secondary education, around ages 9-12, with the final assessment of math anxiety at age 18 (thus the effective age span tested was age 9 through 18). The authors seem somewhat concerned that their results show rather modest effect after analyzing the data, possibly because other, similar studies have sometimes shown stronger effects. I would suggest that with their more adequate sample size and more thorough age sampling range, they are probably providing a more accurate estimate of the true effect size under consideration.

I found the overall coverage of the manuscript to be quite adequate, including the thoroughness of the literature review, the explication of the analysis strategy, and the interpretation of results. The results are entirely sensible, the limitations of the study are portrayed clearly, and need not be expanded in my view. It is a strong paper, one that will contribute to our understanding of math anxiety.

Review form: Reviewer 2 (Irene Mammarella)**Is the manuscript scientifically sound in its present form?**

No

Are the interpretations and conclusions justified by the results?

Yes

Is the language acceptable?

Yes

Do you have any ethical concerns with this paper?

No

Have you any concerns about statistical analyses in this paper?

Yes

Recommendation?

Major revision is needed (please make suggestions in comments)

Comments to the Author(s)

Comments to the manuscript:

Predicting Maths Anxiety From Mathematical Achievement Across the Transition From Primary-to Secondary-Education

Summary:

The present manuscript analyzed a sample of 1104 participants in which math attainment and emotional difficulties were measured at ages 9 and 12, while math and general anxiety were measured at age 18. The main aim of the study was to understand whether math and emotional changes predicted math anxiety in older participants, by also controlling for verbal intelligence, SES, gender and general anxiety.

General remarks:

The manuscript is interesting and clearly written. The strength of the study is the high number of participants and the presence of some variables tested longitudinally. However, I have some concerns which are summarized here below.

Major points:

1. The introduction is well structured and clear. However, the following papers should be considered. At p. 2 when the Authors reported the estimates of math anxiety I suggest considering the paper of Devine, Hill, Carey, & Szucs, 2018 who tested a large sample of children in UK looking for differences between math anxiety levels and math difficulties. Moreover, on the same page, the most recent meta-analysis of Namkung and Peng, 2019 on school-aged students should be mentioned.
2. In the results section, I suggest to delete Table 2, or to move it in the supplementary materials. Descriptive statistics are reported, however, the correlations among variables should be added to have an overall view of the data. Moreover, the linear models are difficult to read and interpret and the suggestion to transform the b values in terms of changes in math anxiety is correct but convoluted. I suggest to report standardized measures in the tables in order to simplify the interpretation of the results. Although many participants were included in the model, math change had a t -value of 1.96, with a $p=.05$ so it seems to me that the relation with math anxiety was not very high, as the Authors admitted. Anyway, with standardized measures, the interpretation could be easier. Finally, in the models reported in Tables 4 and 5, it is not clear why degrees of freedom changed so much. It can depend on the number of participants, of course, who changed across measures, but it is not clear whether measures were added one after the other or all together.
3. The Authors should clarify why also Math pre-transition and SDQ pre-transition were entered into the models. In my view changes in these measures could contribute to explain math anxiety at age 18, but I'm not sure to have well understood the hypothesis related to those measures at age 9.
4. In the limits of the study, the Authors should add that their math attainment derived from teachers' ratings but not from objective measures.

Decision letter (RSOS-191459.R0)

17-Oct-2019

Dear Professor Field

On behalf of the Editors, I am pleased to inform you that your Manuscript RSOS-191459 entitled "Predicting Maths Anxiety From Mathematical Achievement Across the Transition From

Primary- to Secondary-Education" has been accepted for publication in Royal Society Open Science subject to minor revision in accordance with the referee suggestions. Please find the referees' comments at the end of this email.

The reviewers and handling editors have recommended publication, but also suggest some minor revisions to your manuscript. Therefore, I invite you to respond to the comments and revise your manuscript.

- Ethics statement

- Data accessibility

If you wish to submit your supporting data or code to Dryad (<http://datadryad.org/>), or modify your current submission to dryad, please use the following link:
<http://datadryad.org/submit?journalID=RSOS&manu=RSOS-191459>

- Competing interests

- Authors' contributions

- Acknowledgements

- Funding statement

Because the schedule for publication is very tight, it is a condition of publication that you submit the revised version of your manuscript before 26-Oct-2019. Please note that the revision deadline will expire at 00.00am on this date. If you do not think you will be able to meet this date please let me know immediately.

Supplementary files will be published alongside the paper on the journal website and posted on the online figshare repository (<https://rs.figshare.com/>). The heading and legend provided for each supplementary file during the submission process will be used to create the figshare page,

so please ensure these are accurate and informative so that your files can be found in searches. Files on figshare will be made available approximately one week before the accompanying article so that the supplementary material can be attributed a unique DOI.

Kind regards,
Anita Kristiansen
Editorial Coordinator
Royal Society Open Science
openscience@royalsociety.org

on behalf of Dr Emma Hayiou-Thomas (Associate Editor) and Essi Viding (Subject Editor)
openscience@royalsociety.org

Associate Editor Comments to Author (Dr Emma Hayiou-Thomas):

Associate Editor:

Comments to the Author:

Both reviewers and I are in agreement that this paper addresses an important and current topic, is clearly written, and has several notable methodological strengths, specifically the longitudinal design, the large sample size, and the analytical approach. There are nonetheless some relatively minor changes which would further improve the paper, and Reviewer 2 makes several constructive suggestions which will need to be addressed. In particular, please include the correlations among predictors and with outcome variables. Also, although I appreciate your efforts to interpret the magnitude of the effects, this could be made clearer - please consider Reviewer 2's suggestion of including standard scores. Finally, please clarify the concluding statements on p. 10 (lines 12-15), which currently come across as contradictory statements about whether or not the results support the reciprocal model of maths anxiety.

Reviewer comments to Author:

Reviewer: 1

Comments to the Author(s)

A handful of recent papers have examined the issue of which construct, math anxiety or math achievement, leads to or influences the other, in an attempt to gain some evidence on causality on this knotty question. As the authors note, three theories have been articulated - the Debilitating Anxiety Model, the Deficit Model, and the Reciprocal Model - but all suffer from the same

limitations, that the data brought to bear on the issue necessarily come from correlations, hence cannot support causal interpretations. The recent papers, on the other hand, have attempted to gain some leverage on the issue by examining patterns of influence (again, largely the magnitudes of correlations) in longitudinal designs, asking for example if poorer performance at time 1 leads to worse anxiety at time 2 or, instead whether worse anxiety at time 1 leads to worse performance at time 2.

The current manuscript examines the same kind of evidence, but I believe is considerably stronger than the recent papers I've seen, for a variety of reasons. First, it presents data from a far larger sample, well over 1000 cases, in contrast to the 100 or 200 cases normally tested. Second, it spans a considerably longer time range to examine the longitudinal effect; here the range of interest is prior to the transition from primary to secondary education, around ages 9-12, with the final assessment of math anxiety at age 18 (thus the effective age span tested was age 9 through 18). The authors seem somewhat concerned that their results show rather modest effect after analyzing the data, possibly because other, similar studies have sometimes shown stronger effects. I would suggest that with their more adequate sample size and more thorough age sampling range, they are probably providing a more accurate estimate of the true effect size under consideration.

I found the overall coverage of the manuscript to be quite adequate, including the thoroughness of the literature review, the explication of the analysis strategy, and the interpretation of results. The results are entirely sensible, the limitations of the study are portrayed clearly, and need not be expanded in my view. It is a strong paper, one that will contribute to our understanding of math anxiety.

Reviewer: 2

Comments to the Author(s)

Comments to the manuscript:

Predicting Maths Anxiety From Mathematical Achievement Across the Transition From Primary- to Secondary-Education

Summary:

The present manuscript analyzed a sample of 1104 participants in which math attainment and emotional difficulties were measured at ages 9 and 12, while math and general anxiety were measured at age 18. The main aim of the study was to understand whether math and emotional changes predicted math anxiety in older participants, by also controlling for verbal intelligence, SES, gender and general anxiety.

General remarks:

The manuscript is interesting and clearly written. The strength of the study is the high number of participants and the presence of some variables tested longitudinally. However, I have some concerns which are summarized here below.

Major points:

1. The introduction is well structured and clear. However, the following papers should be considered. At p. 2 when the Authors reported the estimates of math anxiety I suggest considering the paper of Devine, Hill, Carey, & Szucs, 2018 who tested a large sample of children in UK looking for differences between math anxiety levels and math difficulties. Moreover, on the same page, the most recent meta-analysis of Namkung and Peng, 2019 on school-aged students should be mentioned.
2. In the results section, I suggest to delete Table 2, or to move it in the supplementary materials. Descriptive statics are reported, however, the correlations among variables should be added to have an overall view of the data. Moreover, the linear models are difficult to read and interpret

and the suggestion to transform the b values in terms of changes in math anxiety is correct but convoluted. I suggest to report standardized measures in the tables in order to simplify the interpretation of the results. Although many participants were included in the model, math change had a t-value of 1.96, with a $p=.05$ so it seems to me that the relation with math anxiety was not very high, as the Authors admitted. Anyway, with standardized measures, the interpretation could be easier. Finally, in the models reported in Tables 4 and 5, it is not clear why degrees of freedom changed so much. It can depend on the number of participants, of course, who changed across measures, but it is not clear whether measures were added one after the other or all together.

3. The Authors should clarify why also Math pre-transition and SDQ pre-transition were entered into the models. In my view changes in these measures could contribute to explain math anxiety at age 18, but I'm not sure to have well understood the hypothesis related to those measures at age 9.

4. In the limits of the study, the Authors should add that their math attainment derived from teachers' ratings but not from objective measures.

Author's Response to Decision Letter for (RSOS-191459.R0)

See Appendix A.

Decision letter (RSOS-191459.R1)

31-Oct-2019

Dear Professor Field,

I am pleased to inform you that your manuscript entitled "Predicting Maths Anxiety From Mathematical Achievement Across the Transition From Primary- to Secondary-Education" is now accepted for publication in Royal Society Open Science.

Kind regards,
Lianne Parkhouse
Editorial Coordinator
Royal Society Open Science
openscience@royalsociety.org

on behalf of Dr Emma Hayiou-Thomas (Associate Editor) and Essi Viding (Subject Editor)
openscience@royalsociety.org

Appendix A

Author response to reviews of

Manuscript RSOS-191459

Predicting Maths Anxiety From Mathematical Achievement Across the Transition From Primary- to Secondary-Education

submitted to *Royal Society Open Science*

RC: *Reviewer Comment* AR: Author Response Manuscript text

Dear Dr Hayiou-Thomas,

Thank you very much for taking the time to consider our revised manuscript for publication at *Royal Society Open Science*. We are delighted that you have accepted the manuscript with only minor revisions. Below we describe those revisions.

1. Editor

RC: Finally, please clarify the concluding statements on p. 10 (lines 12-15), which currently come across as contradictory statements about whether or not the results support the reciprocal model of maths anxiety.

AR: We have rephrased these sentences as:

Nonetheless, small though the effect is, it is consistent with the wider literature (Maloney, Ansari, and Fugelsang 2011; Maloney et al. 2010; Núñez-Peña and Suárez-Pellicioni 2014; Dowker, Sarkar, and Looi 2016; Hembree 1990; Ma and Xu 2004; Wu et al. 2012). In this sense, the results support the hypothesis from the reciprocal and deficit models of maths anxiety that poor performance leads to increased maths anxiety, but also imply that this relationship is very weak.

2. Reviewer #1

RC: The current manuscript examines the same kind of evidence, but I believe is considerably stronger than the recent papers I've seen, for a variety of reasons. First, it presents data from a far larger sample, well over 1000 cases, in contrast to the 100 or 200 cases normally tested. Second, it spans a considerably longer time range to examine the longitudinal effect; here the range of interest is prior to the transition from primary to secondary education, around ages 9-12, with the final assessment of math anxiety at age 18 (thus the effective age span tested was age 9 through 18). The authors seem somewhat concerned that their results show rather modest effect after analyzing the data, possibly because other, similar studies have sometimes shown stronger effects. I would suggest that with their more adequate sample size and more thorough age sampling range, they are probably providing a

more accurate estimate of the true effect size under consideration. I found the overall coverage of the manuscript to be quite adequate, including the thoroughness of the literature review, the explication of the analysis strategy, and the interpretation of results. The results are entirely sensible, the limitations of the study are portrayed clearly, and need not be expanded in my view. It is a strong paper, one that will contribute to our understanding of math anxiety.

AR: We thank Reviewer 1 for the very positive comments. There are no issues to be addressed from this review.

3. Reviewer #2

RC: The introduction is well structured and clear. However, the following papers should be considered. At p. 2 when the Authors reported the estimates of math anxiety I suggest considering the paper of Devine, Hill, Carey, & Szucs, 2018 who tested a large sample of children in UK looking for differences between math anxiety levels and math difficulties. Moreover, on the same page, the most recent meta-analysis of Namkung and Peng, 2019 on school-aged students should be mentioned.

AR: Thanks for these suggestions. We have added both of these references to page 2. Specifically, the Devine et al. study is referenced as follows:

... Chinn reported estimates of 2-6% for high maths anxiety in secondary school pupils in the UK (Chinn 2009). Similarly, a large study of 1757 UK school children between years 4 (age 8-9) and 7 (age 12-13) found that 11% of children had an average score above 'moderate anxiety' on a maths anxiety scale (Devine et al. 2018). Researchers investigating less severe levels of maths anxiety (mostly in students) have tended to report higher estimates, between 11% to 68% (Ashcraft and Moore 2009; Betz 1980; Johnston-Wilder, Brindley, and Dent 2014; Richardson and Suinn 1972). Regardless of the variability in these estimates, there are evidently substantial number of individuals experiencing varying levels of maths anxiety, and its corresponding negative impact.

The Namkung and Peng (2019) study has been included at the point where we discuss previous meta-analyses:

Two meta-analyses report the correlation between maths anxiety and maths performance to be between $r = -.27$ to $-.36$ in high school and college students (Hembree 1990; Ma 1999). The most recent meta-analysis reports the overall correlation as $r = -.27 [-.31, -.23]$ in primary school children and $r = -.36 [-.40, -.32]$ in secondary school children (Namkung, Peng, and Lin 2019).

RC: In the results section, I suggest to delete Table 2, or to move it in the supplementary materials. Descriptive statics are reported, however, the correlations among variables should be added to have an overall view of the data.

AR: The former Table 2 (containing patterns of missing data) has been moved to the supplementary materials. The sentence referring to them has been changed to:

Within this sample there were missing values across some of the predictors. There were 952 complete cases and the patterns of missing data are shown in the supplementary information.

The Table 2 in this revision is a table of correlations for the variables in the models as requested.

RC: Moreover, the linear models are difficult to read and interpret and the suggestion to transform the b values in terms of changes in math anxiety is correct but convoluted. I suggest to report standardized measures in the tables in order to simplify the interpretation of the results. Although many participants were included in the model, math change had a t-value of 1.96, with a p=.05 so it seems to me that the relation with math anxiety was not very high, as the Authors admitted. Anyway, with standardized measures, the interpretation could be easier. Finally, in the models reported in Tables 4 and 5, it is not clear why degrees of freedom changed so much. It can depend on the number of participants, of course, who changed across measures, but it is not clear whether measures were added one after the other or all together.}

AR: I respectfully disagree with the suggestion that standardized parameter estimates simplify the interpretation. I think this is largely a matter of opinion rather than anyone being correct. Raw coefficients have the benefit of retaining the original scale of measurement. In the current paper, I believe this makes it easier to see the 'real world' effect of the predictors. There is a clear mapping between a unit change on a predictor and the corresponding change in maths anxiety. What you lose, is the ability to compare predictors (i.e. see the relative effect of one predictor vs another). Baguley (2009) has convincingly (in my opinion) argued that raw effect sizes should always be reported and rarely is the standardized effect size more helpful than the unstandardised one (<https://pdfs.semanticscholar.org/86b6/bef80331f6afbbcb7371bd23ab3abc3ba0b2.pdf>). However, I also want to be responsive to the reviewer. My compromise has been to reproduce Tables 4 and 5 from the main paper in the Supplementary materials but with values deriving from models in which predictors and outcomes were standardized (in other words, the parameter estimates are standardized). This information gives readers the best of both worlds - they can follow the arguments in the paper, but refer to standardized coefficients in the additional materials if they find that helpful to aid their understanding. I hope this is a reasonable compromise.

The MS text has been amended to refer to these tables:

Table 4 shows the model parameters for predictors of maths anxiety at age 18 (an equivalent table reporting the same model fitted to standardized scores can be found in the supplementary information).

Table 5 shows an unplanned, *post hoc*, model that quantifies the degree to which biological sex moderated the effects on maths anxiety of maths attainment and SDQ emotional symptoms prior-to and across the school transition. Biological sex did not significantly moderate any of the effects of maths attainment of emotional symptoms on subsequent maths anxiety. An equivalent table reporting the same model fitted to standardized scores can be found in the supplementary information.

The degrees of freedom in the models are a function of the multiple imputation procedure, which is why the values seem strange. I have clarified that all predictors were entered simultaneously in the Data Analysis Plan section of the paper:

In phase two, the main model was simply a linear model predicting maths anxiety at age 18 from SES, biological sex, verbal attainment, general anxiety at age 18 and the four variables from the previous models: SDQ emotional symptoms (pre-transition), SDQ emotional symptoms (change), Maths attainment (pre-transition), and Maths attainment (change). All predictors were entered simultaneously. Table 2 shows the correlations between these variables.

RC: The Authors should clarify why also Math pre-transition and SDQ pre-transition were entered into the models. In my view changes in these measures could contribute to explain math anxiety at age 18, but I'm not sure to have well understood the hypothesis related to those measures at age 9.

AR: We have clarified their inclusion at the end of the introduction:

With respect to the school environment the transition from primary-to-secondary education represents a significant change in the environment that has a negative impact on many children. In addition, the impact that it has is likely to be worse for children high on trait anxiety. Specifically, it might heighten or maintain high levels of trait anxiety. Therefore, trait anxiety before and after the transition from primary-to-secondary education should also predict future maths anxiety.

RC: In the limits of the study, the Authors should add that their math attainment derived from teachers' ratings but not from objective measures.

AR: We added this to the limitations:

Finally, although the measure of Maths attainment derived from teachers were based on the National Curriculum examination benchmarks, they nevertheless reflect subjective ratings of the children's ability and are not as objective as, for example, National Curriculum exam scores.

We hope that you find these revisions sufficient to accept the paper and look forward to hearing from you in due course. Yours sincerely

Andy Field

4. References

- Ashcraft, Mark H., and Alex M. Moore. 2009. "Mathematics Anxiety and the Affective Drop in Performance." *Journal of Psychoeducational Assessment* 27: 197–205. <https://doi.org/10.1177/0734282908330580>.
- Betz, Nancy E. 1980. "Prevalence, Distribution, and Correlates of Math Anxiety in College Students." *Journal of Counseling Psychology* 25: 441–48. <https://doi.org/10.1037/0022-0167.25.5.441>.
- Chinn, Steve. 2009. "Mathematics Anxiety in Secondary Students in England." *Dyslexia* 15: 61–68. <https://doi.org/10.1002/dys.381>.
- Devine, Amy, Francesca Hill, Emma Carey, and Dénes Szücs. 2018. "Cognitive and Emotional Math Problems Largely Dissociate: Prevalence of Developmental Dyscalculia and Mathematics Anxiety." *Journal of Educational Psychology* 110 (April): 431–44. <https://doi.org/http://dx.doi.org/10.1037/edu0000222>.
- Dowker, Ann, Amar Sarkar, and Chung Yen Looi. 2016. "Mathematics Anxiety: What Have We Learned in 60 Years?" *Frontiers in Psychology* 7. <https://doi.org/10.3389/fpsyg.2016.00508>.

- Hembree, Ray. 1990. "The Nature, Effects, and Relief of Mathematics Anxiety." *Journal for Research in Mathematics Education* 21: 33–46.
- Johnston-Wilder, Sue, Janine Brindley, and Philip Dent. 2014. *A Survey of Mathematics Anxiety and Mathematical Resilience Among Existing Apprentices*. London: Gatsby Charitable Foundation. <http://wrap.warwick.ac.uk/73857/>.
- Ma, X. 1999. "A Meta-Analysis of the Relationship Between Anxiety Toward Mathematics and Achievement in Mathematics." *Journal for Research in Mathematics Education* 30: 520–40. <https://doi.org/10.2307/749772>.
- Ma, X, and Jiangmin Xu. 2004. "The Causal Ordering of Mathematics Anxiety and Mathematics Achievement: A Longitudinal Panel Analysis." *Journal of Adolescence* 27: 165–79. <https://doi.org/10.1016/j.j.adolescence.2003.11.003>.
- Maloney, Erin A., Daniel Ansari, and Jonathan A. Fugelsang. 2011. "The Effect of Mathematics Anxiety on the Processing of Numerical Magnitude." *Quarterly Journal of Experimental Psychology* 64: 10–16. <https://doi.org/10.1080/17470218.2010.533278>.
- Maloney, Erin A., Evan F. Risko, Daniel Ansari, and Jonathan Fugelsang. 2010. "Mathematics Anxiety Affects Counting but Not Subitizing During Visual Enumeration." *Cognition* 114: 293–97. <https://doi.org/10.1016/j.cognition.2009.09.013>.
- Namkung, Jessica M., Peng Peng, and Xin Lin. 2019. "The Relation Between Mathematics Anxiety and Mathematics Performance Among School-Aged Students: A Meta-Analysis." *Review of Educational Research* 89 (June): 459–96. <https://doi.org/10.3102/0034654319843494>.
- Núñez-Peña, María I., and Macarena Suárez-Pellicioni. 2014. "Less Precise Representation of Numerical Magnitude in High Math-Anxious Individuals: An ERP Study of the Size and Distance Effects." *Biological Psychology* 103: 176–83. <https://doi.org/10.1016/j.biopsycho.2014.09.004>.
- Richardson, Frank C., and Richard M. Suinn. 1972. "Mathematics Anxiety Rating Scale - Psychometric Data." *Journal of Counseling Psychology* 19: 551. <https://doi.org/10.1037/h0033456>.
- Wu, Sarah, Hitha Amin, Maria Barth, Vanessa Malcarne, and Vinod Menon. 2012. "Math Anxiety in Second and Third Graders and Its Relation to Mathematics Achievement." *Frontiers in Psychology* 3. <https://doi.org/10.3389/fpsyg.2012.00162>.